# Impacts of reductions in anthropogenic aerosols and greenhouse gases toward carbon neutrality on dust pollution over the Northern Hemisphere dust belt

Shicheng Yan[1,2], Yang Yang[1,2*], Lili Ren[3], Hailong Wang[4], Pinya Wang[2], Lei Chen[2], Jianbin Jin[1,2], Hong Liao[1,2]

[1]State Key Laboratory of Climate System Prediction and Risk Management/Jiangsu Key Laboratory of Atmospheric Environment Monitoring and Pollution Control/Jiangsu Collaborative Innovation Center of Atmospheric Environment and Equipment Technology/Joint International Research Laboratory of Climate and Environment Change, Nanjing University of Information Science and Technology, Nanjing, Jiangsu, China
[2]School of Environmental Science and Engineering, Nanjing University of Information Science and Technology, Nanjing, Jiangsu, China
[3]School of Environment and Ecology, Jiangsu Open University, Nanjing, Jiangsu, China
[4]Atmospheric, Climate, and Earth Sciences Division, Pacific Northwest National Laboratory, Richland, Washington, USA

* Correspondence to Yang Yang (yang.yang@nuist.edu.cn)

**Abstract**

To mitigate future global warming, many countries have implemented rigorous climate policies for carbon neutrality. Given some shared emission sources with greenhouse gases (GHGs), aerosol particles and their precursor emissions are expected to be reduced as the consequences of global efforts in climate mitigation and environmental improvement, potentially inducing complex climate feedbacks. However, a clear understanding of the individual effects of anthropogenic aerosols and GHGs on natural dust concentrations has not yet emerged, especially in the carbon neutral scenario. Here, we assess the large-scale impacts of reductions in anthropogenic GHGs and aerosol under a carbon neutral scenario in 2060 on natural dust emissions and concentrations over the low- to mid-latitudes in the Northern Hemisphere using the fully coupled Community Earth System Model. Our findings demonstrate a decline in atmospheric dust loading toward carbon neutrality (SSP1-1.9) relative to the high fossil fuel scenario (SSP5-8.5). Mechanistic analysis reveals counteracting modulation mechanisms: (i) Reductions in aerosols amplify surface downwelling shortwave radiation, convection and wind speed, thereby promoting dust emissions by 6–12% and concentrations by 4–20% over North Africa, the Central Asia Desert and East Asia; (ii) GHGs reductions diminish the land-ocean thermal contrast and wind speed, suppressing dust emissions by 6–15% and concentrations by 8–20% mainly over the Central Asia Desert and North Africa. The latter drives the future dust responses. These results highlight that carbon neutral strategies not only achieve climate mitigation goals and air quality improvements, but also generate synergistic benefits through dust pollution suppression.

## 1. Introduction

Dust aerosols are a crucial component of the Earth-atmosphere system, exerting multifaceted influences on environment and climate (Chen et al., 2024; Hu et al., 2023). They play a significant role in modulating the Earth's radiation budget via aerosol-cloud and aerosol-radiation interactions. Dust aerosols absorb longwave radiation and scatter shortwave radiation, thereby influencing atmospheric radiative balance and surface energy fluxes (Kok et al., 2017, 2023; Liu et al., 2021). Additionally, dust aerosols act as cloud condensation nuclei, modifying cloud microphysical properties and subsequently affecting cloud development and precipitation patterns (Min et al., 2009; Yuan et al., 2021; Zhang et al., 2021). In addition, mineral dust transports iron to marine ecosystems, stimulating phytoplankton growth and enhancing carbon fixation (Jickells et al., 2005; Pabortsava et al., 2017). Furthermore, dust can reduce visibility, degrade air quality and have important impacts on public health, particularly in arid and semiarid regions (Fussell et al., 2021; Goudie et al., 2014; Li et al., 2024; Roy et al., 2023). These health risks are extended beyond proximal desert margins to distal urban centers by intercontinental transport mechanisms (Griffin et al., 2007; Meng et al., 2023).

The global primary sources of dust emissions are located in the arid zones of the low- to mid-latitudes in the Northern Hemisphere, with core areas concentrated in the Sahara Desert of North Africa, the Central Asia Desert, Arabian Desert, Taklamakan Desert, and Gobi Desert of East Asia, which is often called the dust belt (Prospero et al., 2002; Shao et al., 2011). Specifically, the North African desert, as the world's largest dust source, injects approximately 1.0-1.5 billion tons of dust aerosols annually into the atmosphere, accounting for 50%-65% of the global total dust emissions (Tanaka et al., 2006; Ginoux et al., 2004). Meanwhile, Asian dust sources contribute 30%-40% of the global dust flux and are identified as the second-largest emission center (Kok et al., 2021).

Dust emission is influenced by climate change, determined by a combination of natural and anthropogenic factors, including greenhouse gases (GHGs) concentrations, aerosol loading, and land use, with anthropogenic contributions exhibiting increasing influence in the post-industrial era (Gui et al., 2022; Tegen et al., 2004). Variations in GHGs concentrations further regulate dust transport through large-scale atmospheric teleconnections. Elevated GHGs levels amplified the North Atlantic Oscillation (NAO)

(Kuzmina et al., 2005), which changed atmospheric circulation patterns and enhanced dust advection to South Asia (Banerjee et al., 2021). The strengthened West African monsoon under warming conditions was found to amplify dust emissions (Wubben et al., 2024). In the arid and semi-arid regions of North and Central Asia, surface warming enhanced atmospheric instability, thereby intensifying vertical convective motions and significantly increasing dust emission fluxes (Zhou et al., 2023). Anthropogenic aerosols are recognized as an important forcing factor in global and regional climate systems (Ramanathan et al., 2001; Myhre et al., 2017). Analyses of observations from 1979 to 2013 showed that anthropogenic sulfate aerosols over the Asian monsoon region suppressed dust emissions in East Asia by altering atmospheric dynamics (Xie et al., 2025). Specifically, sulfate-induced shifts in the Asian westerly jet enhanced precipitation and reduced surface wind speeds across arid and semi-arid source regions, thereby limiting dust mobilization. Model simulations illustrated that the combined reduction of carbonaceous aerosols (black carbon and organic carbon) and increased sulfate emissions in South Asia synergistically caused atmospheric cooling over continental regions, which attenuated the zonal thermal gradient, resulting in a weakening of the Indian summer monsoon circulation (Das et al., 2020). Concurrently, this altered atmospheric circulation suppressed dust emissions from the Arabian Peninsula and inhibited dust transport across the Arabian Sea. Observational and reanalysis data from the COVID-19 pandemic period revealed that anthropogenic aerosol emission reductions over the Indian subcontinent amplified the Indian summer monsoon intensity and triggered anomalous convective activity over the tropical Indian Ocean, which increased surface wind speeds and enhanced dust lifting over the Arabian Peninsula (Francis et al., 2022). Modeling studies have found that reductions in anthropogenic aerosol emissions along the West African coast led to a decrease in aerosol loading, triggering a northward shift of the monsoonal precipitation belt. This meridional displacement subsequently enhanced surface wind speeds over the Saharan arid zone, thereby increasing mineral dust emission fluxes through intensified wind erosion processes (Menut et al., 2019).

Under future climate change, dust distribution will vary depending on the projected scenarios. Using the Coupled Model Intercomparison Project Phase 5 (CMIP5) multi-model simulations, Singh et al. (2017) showed a 30% increase in regional dust loading over the South Asian monsoon region by the end of the 21st century (2076-2100) relative to 1976-2000 under the RCP8.5 scenario. Zhao et al.

(2023) analyzed the multi-model results under four Shared Socioeconomic Pathways (SSPs) from the Coupled Model Intercomparison Project Phase 6 (CMIP6) and found that global dust loading was expected to increase by 2.0-12.5% by the end of the 21st century in most future scenarios, except for SSP3-7.0, which shows a slight decline. Liu et al. (2024) estimated a substantial increase in dust mass loading over North Africa during 2081-2100 under SSP1-2.6, SSP2-4.5, SSP3-7.0, and SSP5-8.5 scenarios from bias-corrected CMIP6 models. Woodward et al. (2005) showed through HadCM3-coupled model experiments that the annual mean global dust burden would rise by 225%, from the 2000 baseline ($4\times10^4$ mg m$^{-2}$) to $1.3\times10^5$ mg m$^{-2}$ by 2100, under a medium-emission scenario, attributed to desertification and climate change. Gomez et al. (2023) projected that rising $CO_2$ concentrations would elevate global mean $PM_{2.5}$ levels, partly driven by intensified dust aerosol emissions attributable to a strengthened West African monsoon. Akinsanola et al. (2025) found that African easterly wave activity was projected to undergo a robust intensification across the Sahel region under both SSP2-4.5 and SSP5-8.5 scenarios by the end of the 21st century, with profound implications for Saharan dust emission and transport. These studies mainly focus on investigating dust variations under different Shared Socioeconomic Pathways, thereby examining only the combined effects of anthropogenic aerosols and GHGs. However, relatively little attention has been paid to quantifying the individual contributions of anthropogenic aerosols and GHGs changes to the changing dust concentrations in the future, especially in the carbon-neutral scenario.

The future climate changes toward carbon neutrality would also affect dust aerosols, which remains largely unknown. Many countries have committed to achieve carbon neutrality by the middle of the 21st century to limit global temperature rise to below 2°C or even 1.5°C by the end of the 21st century. The pursuit of carbon neutrality will reshape anthropogenic emissions associated with climate and environmental policies, driving changes in atmospheric composition and radiative forcing (Wang et al., 2023; Yang et al., 2023). As nations reduce GHGs and aerosol emissions to mitigate global warming, these shifts are expected to induce complex climate influences. Studies have suggested that anthropogenic aerosol reductions could enhance surface downwelling shortwave radiation, elevate near-surface temperatures, and increase wind speed (Lei et al., 2023; Ren et al., 2024). Projections indicated that by the end of the 21st century, interannual precipitation variability will intensify by 3.9% and 5.3% under 1.5°C and 2.0°C warming scenarios, respectively (Chen et al., 2020). Consequently, the

implementation of carbon neutrality policies is likely to modify the current climate state and affect various meteorological variables (Seager et al., 2019; Lee et al., 2013), which are expected to influence dust mobilization.

In the carbon-neutral future, reductions in GHGs and aerosols can change climate and meteorological factors, which further affect dust emissions and concentrations. However, existing studies typically focus on dust flux responses to climate change under future scenarios, thereby examining only the combined effects of anthropogenic aerosols and GHGs, which also have yet to quantify dust response to future climate change due to individual changes in anthropogenic aerosols and GHGs for pursuing carbon neutrality goals (Zhao et al., 2023; Liu et al., 2024). In this study, we conduct Earth system model experiments to assess the impact of aerosols and GHGs reductions toward carbon neutrality on meteorological variables such as precipitation, relative humidity, and wind speed, as well as their implications for dust emissions and concentrations. Although dust is from both natural and anthropogenic sources. This study only focuses on dust from natural sources without considering anthropogenic dust. Given that the combined contribution of dust sources from the North Africa and Asia exceeds 80% of global dust emissions, this study strategically focuses on the dust belt regions, including the Sahara Desert, Central Asia Desert, Arabian Desert, Taklamakan Desert, and Gobi Desert. The findings of this study aim to provide valuable insights to guide the establishment of dust prevention measures and strategies in global pursuit of carbon neutrality. The paper is structured as follows. The method and data are presented in Sect. 2. The results of dust changes related to the reductions in GHGs and aerosols are shown in Sect. 3. The discussion and the conclusions are given in Sect. 4.

## 2. Methods

## 2.1 Model Description

The fully coupled Community Earth System Model version 1.2.2 (CESM1) (Hurrell et al., 2013) is used to investigate the effects of meteorological changes induced by anthropogenic aerosols and GHGs under carbon neutrality on dust emissions and concentrations. The atmospheric component utilizes the Community Atmosphere Model version 5 (CAM5), which simulates the major aerosol species, including sulfate, black carbon, primary organic aerosol, secondary organic aerosol, mineral dust and sea salt. These aerosols are distributed in the four lognormal size

distribution modes (i.e., Aitken, accumulation, coarse, and primary carbon modes) (Liu
et al., 2016). Simulations are conducted at 1.9°×2.5° horizontal resolution with 30
vertical layers. Aerosol particles within the same mode are mixed internally, whereas
external mixing assumption is treated for particles between different modes. The dust
emission flux is calculated using the Dust Entrainment and Deposition model
developed by Zender et al. (2003), which is implemented in the Community Land
Model version 4 (CLM4; Oleson et al., 2010). Dust particles are divided into four bins
(0.1-1.0, 1.0-2.5, 2.5-5.0, and 5.0-10.0 μm) in CLM4, and subsequently redistributed to
four modes of the Modal Aerosol Module scheme. The emission or mobilization
process is governed by the synergistic effects of multiple controlling parameters,
including wind friction speed, vegetation cover, and surface soil moisture content.
Aerosol direct and indirect radiative effects are incorporated in CAM5 (Ma et al., 2022).
Furthermore, optimized parameterization schemes for key aerosol processes in CAM5,
such as convective transport and wet deposition, have been implemented to enhance
model performance (Wang et al., 2013). The dynamic oceanic component in CESM1
uses the Parallel Ocean Program version 2 (POP2). In this study, emissions of aerosols
and precursors and GHGs concentrations are obtained from the CMIP6 input data,
specifically adopting the SSP1-1.9 and SSP5-8.5 (shared socioeconomic pathways).
Future emission inventories build on the Shared Socioeconomic Pathways, providing
standardized multidimensional parameters (e.g., population, economy, technology,
environment, institutions) and qualitative narratives at national/regional scales (van
Vuuren et al., 2017; Kriegler et al., 2017; Fujimori et al., 2017; Calvin et al., 2017;
Fricko et al., 2017).

## 207    2.2 Experimental Design

To quantify the impacts of anthropogenic aerosols and GHGs on future dust
toward carbon neutrality, four sets of CESM1 equilibrium simulations are designed,
comprising one baseline (Fut_SSP585) and three sensitivity experiments
(Fut_CNeutral, AA_CNeutral and GHG_CNeutral). The SSP1-1.9 represents a
sustainable development scenario focused on ecological restoration, conservation, and
a significant reduction in fossil fuel dependence. This pathway is considered the most
likely to achieve the 1.5 °C target under the Paris Agreement and carbon neutrality in
the mid-21st century (Su et al., 2021; Wang et al., 2023; Zhu et al., 2024). In contrast,
the SSP5-8.5 follows a high fossil fuel consumption with substantial associated
emissions (Meinshausen et al., 2020). Many countries had committed to achieving
carbon neutrality by 2050 or 2060, with most targets set for the post-2050 period (Chen
et al., 2022). Focusing on the year 2060 therefore ensures direct alignment with policy
timelines and enhances the practical relevance of our results.
The Fut_SSP585 simulation prescribes global GHGs concentrations and
anthropogenic emissions of aerosols and precursors from the CMIP6 input data, with
all forcings held at 2060 levels under the SSP5-8.5 scenario. In Fut_CNeutral
experiment, GHGs concentrations, aerosols, and their precursor emissions are adopted
following SSP1-1.9 emission pathway in 2060, enabling isolation of combined effects
of aerosols and GHGs through comparison with the baseline. The AA_CNeutral
experiment applies anthropogenic emissions of aerosols and precursors from SSP1-1.9
while retaining GHGs concentrations under SSP5-8.5, allowing aerosol effect
quantification by comparing with the baseline. Conversely, we also perform the
GHG_CNeutral simulations in which GHGs concentrations are set to the 2060 levels
under SSP1-1.9, along with aerosol emissions using SSP5-8.5 input data, which allows
comparison with the baseline to estimate the climate impacts of GHGs. One additional
experiment, Fut_2020, is also performed for the model evaluation, with GHGs
concentrations and aerosol emissions set to the 2020 levels under SSP1-1.9. All
simulations are initialized with the same conditions and only the GHGs concentrations
and/or aerosol emissions change in time and space every month. All experiments are
conducted with three ensemble members of different initial conditions, achieved by
applying a small initial perturbation to atmospheric temperature. Each ensemble
member is run for 100 years, with the initial 40 years considered as model spin-up
period, retaining the latter 60 years for analysis.
**2.3 Model Evaluation**
Numerous studies documented the hemispheric asymmetry of global dust sources,
with most emissions originated from northern hemisphere arid zones, notably North
Africa, Central Asia, East Asia, and the Middle East (Shao et al., 2011; Ginoux et al.,
2012; Yang et al., 2022). Consistent with prior studies that highlight peak dust activities
during boreal spring and summer in these regions (Ginoux et al., 2012; Nabavi et al.,
2016; Jethva et al., 2005, Choobari et al, 2014), our seasonal analysis for simulations
in 2060 also reveals substantially elevated dust emissions and concentrations in warm
seasons, especially spring, compared to autumn and winter (Figure 1). In this study, we

mainly focus on spring dust activities. To evaluate model's dust simulation performance, dust optical depth from model results in boreal spring of 2020 is compared with CALIPSO satellite retrievals averaged over 2017–2021. The model reasonably reproduces the overall spatial distribution of dust optical depth (Figure 2), but overestimates dust loading over parts of Central Asia, Eastern Africa and the Gobi Desert. Similar discrepancies have been noted in existing studies, indicating that the deviations between the model and observations are primarily attributable to the topographic source function and the dust emission scheme used in the model (Wu et al., 2020), which could potentially lead to bias in the quantitative analysis of the results.

## 3 Results

## 3.1 Changing dust aerosol toward carbon neutrality

Figures 3a and 3b present the spatial patterns of changes in emission fluxes and near-surface concentrations of dust aerosols between carbon neutrality (SSP1-1.9) and high fossil fuel (SSP5-8.5) scenarios driven by both fixed anthropogenic aerosols and GHGs in 2060. Under the strong decline in anthropogenic emissions toward carbon neutrality, marked reductions in dust emissions (3–12%) and concentrations (4–16%) are observed across primary source regions (Figure 4a-b), particularly the North African dust belt and Central Asian arid corridor, whereas increases in dust emission (3–12%) and concentrations (4–8%) are found over East Asian dust source regions. Dust concentrations in most regions exhibit reductions, exceeding 40 μg m$^{-3}$ over North Africa and Central Asia, while northwestern China and the North China Plain show a weak increase in dust concentrations.

The simulated future changes in dust concentrations are the combined effects of the reduction of anthropogenic aerosols and GHGs. Here we also investigate their respective impacts on future dust changes through sensitivity experiments. Figures 3c-d illustrate the responses of emission fluxes and near-surface concentrations of dust to anthropogenic aerosol reductions in SSP1-1.9 relative to SSP5-8.5, while 3e-f demonstrate the responses to GHGs reduction alone. The future reductions in anthropogenic aerosols would lead to significant increases in dust emissions (6–12%) and concentrations (4–20%) across the dust belt (Figure 4c-d). However, GHGs reduction induces decreases in dust loads mainly over North Africa and Central Asia. These contrasting patterns indicate opposite dust responses to future reductions in anthropogenic aerosols and GHGs. The following sections illustrate possible

mechanisms derived from the analysis of key meteorological drivers and their
association with emission reduction strategies.

## 3.2 Dust increases due to anthropogenic aerosols reductions

Pursuing the carbon neutrality leads to substantial reductions in anthropogenic
emissions of aerosols and precursors. As shown in Figure 5, CMIP6 experiments show
decreases exceeding $8\times10^{-13}$ kg m$^{-2}$ s$^{-1}$ in anthropogenic emissions of aerosols and
precursors, including black carbon, sulfur dioxide and precursor gases of secondary
organic aerosols, over polluted eastern China, South Asia, and parts of Europe and
North Africa in 2060 under SSP1-1.9 scenario compared to SSP5-8.5, while primary
organic matter emissions slightly increase by $4$–$8\times10^{-13}$ kg m$^{-2}$ s$^{-1}$. Although
anthropogenic aerosol emission changes are primarily concentrated in Asia, reductions
in aerosol optical depth (AOD) of approximately 0.01–0.05 are also evident over remote
regions including Northern Africa (Figure 6a), mainly due to the decreases in sulfate
aerosol (Figures 6b and 6c). Along with the aerosol reduction, the surface downwelling
shortwave radiation increases by 4–12W m$^{-2}$ (Figure 7a), which further increases the
land surface temperatures by more than 0.6 ℃ over eastern China, Southeast Asia and
North Africa and 0.9 ℃ over South Asia (Figure 7b). Enhanced convective instability
due to the warmer surface condition elevates planetary boundary layer (PBL) heights
over most land regions (Figure 7c). Furthermore, diminished atmospheric heating from
light-absorbing aerosols (e.g., black carbon) in the air reduces lower tropospheric
stability, intensifying convective conditions and resulting in an increase in the PBL
height. The associated strengthening of vertical exchange processes enhances near-
surface wind speeds by 0.05–0.1 m s$^{-1}$ through downward momentum transfer (Figure
8a) (Qin et al., 2024). Note that, the spatial patterns of changes in PBL height show a
mismatch with dust emission changes in some regions, which arises from the imperfect
correspondence between boundary layer height and surface wind speed and has been
reported in many studies (e.g., Jacobson et al. 2006; Qin et al., 2024). The wind speed
responses to aerosol changes reported in these studies agrees with our findings, and the
mechanistic interpretation that aerosol reduction increases wind speed is also consistent
with their established physical understanding. Related to the surface warming driven
by anthropogenic aerosol reductions, relative humidity and soil water content decrease
(Figures 8b-c). These changes in meteorological and land surface conditions explain
the simulated increases (exceeding $2\times10^{-9}$ kg m$^{-2}$ s$^{-1}$) in dust emissions across the dust

belt due to the anthropogenic aerosol reductions toward carbon neutrality (Figure 3c). This result is consistent with previous studies (Menut et al., 2019; Xie et al., 2025).

Previous studies have established a robust positive correlation between near-surface wind speed and dust emission fluxes, particularly in arid dust source regions characterized by chronically low soil moisture and minimal precipitation inputs (Zender et al., 2003; Dong et al., 2006). Our analysis reveals that anthropogenic aerosol reductions in SSP1-1.9 relative to SSP5-8.5 amplify 10-m wind speed by 0.05–0.10 m s$^{-1}$ across core dust sources (Figures 8a), driving intensified dust emission fluxes (6–12%) and near-surface concentrations (8–16%) in North and Central Africa (Figures 3c-d, Figures 4c-d). The dust-wind speed relationship is modulated by emission thresholds. In arid areas, the threshold of wind speed for dust mobilization increases with rising relative humidity (Ravi et al., 2005). This is primarily due to the enhanced adsorption layer interactions created by overlapping water films on adjacent soil particles (Ravi et al., 2005). Consequently, after the reduction of anthropogenic aerosols, reduced relative humidity by −1% to −3% (Figure 8b) lowers the critical threshold of wind speed, particularly in Central Africa and East Asia. Additionally, in the major dust source regions, precipitation changes are minimal and statistically insignificant (Figure 8d), which do not have a large influence on dust concentrations after emitting into the atmosphere.

## 3.3 Dust decreases due to greenhouse gas reductions

Figure 9a illustrates the surface temperature distribution in 2060 under SSP5-8.5, highlighting persistent land-ocean thermal contrast with continental temperatures around dust source regions much higher than oceanic values. Due to GHGs reductions in SSP1-1.9 relative to SSP5-8.5, surface temperatures decrease by 1.8–3.0 °C over land and 1.2–1.8 °C over adjacent oceans (Figure 9b), where the overall land-sea contrast is largely due to the higher heat capacity of water than land surface. Figures 9c and 9d respectively depict the zonal and meridional distributions of surface temperatures over the Sahara Desert of North Africa. Notably, the surface cooling due to GHGs reductions is stronger over the Sahara Desert (10°–30°N, 10°W–30°E) than that over the Mediterranean Sea (north of 30°N) and North Atlantic Ocean (west of 10°W). It diminishes the land-sea temperature gradient, thereby contributing to the decline in wind speed over North Africa (Figure 10a). Central Asia Desert also demonstrates a stronger temperature reduction than the surrounding Caspian Sea

(Figure 9e) and high latitude regions, weakening the land-sea thermal gradient and thereby driving the decrease in surface wind speed throughout Central Asia (Figure 10a). By reducing the land-ocean thermal contrast, GHG mitigation lowers surface wind speeds over major dust source regions, leading to a consequent decline in dust emissions (exceeding $2 \times 10^{-9}$ kg m$^{-2}$ s$^{-1}$) (Figure 3e), which is consistent with previous study. Qu et al. (2025) studied prolonged wind droughts in a warming climate. Under the SSP5-8.5 scenario, they found that wind droughts decreased in the tropics, primarily due to increased wind speeds. Reversely, in the tropics, global warming amplifies the land-ocean thermal contrast, thereby strengthening winds. Thus, the mechanism of wind speed reduction is consistent with established understanding. As a result of the GHGs reduction implementation, the marked temperature reduction suppresses surface evaporation and alters atmospheric saturation vapor pressure, thereby increasing relative humidity by 1–3% across Northern Hemisphere dust source areas (Figure 10b).

Dust emission suppression in North African and Central Asian regions (Figure 3e) is primarily attributed to the weakened surface wind speeds induced by GHGs reduction (Figure 10a). The GHGs reduction elevates relative humidity (Figure 10b), which raises the critical threshold wind velocity required for dust mobilization. It further reduces dust emission fluxes by 6–15% and atmospheric dust concentrations by 8–20% (Figure 4e-f), particularly in the North African and Central Asian source regions, even though the soil moisture slightly increases in some regions (Figure 10c). This finding is consistent with previous research indicating that dust emissions across most source regions are significantly lower under the low-emission scenarios than under high-emission scenarios (Zhao et al., 2023; Liu et al., 2024; Gomez et al., 2023). The precipitation does not show significant changes over the North Africa and Central Asia (Figure 10d). Over East Asia, the decreases in precipitation and soil water, likely related to the changing atmospheric circulation and moisture transport due to GHGs reductions, slightly promote the dust emissions over some parts of Taklamakan Desert and Gobi Desert (Figure 3e). However, decreases in wind speed do not favor the dust transport (Figure 10a) and are conducive to the local dust deposition. It can be confirmed by the changes in dust deposition that more dust is removed from the atmosphere over the Taklamakan Desert and the downwind North China Plain (Figure 11) and the increase in dust removal surpasses the increase in dust emission ($0.5 \times 10^{-9}$ to $2 \times 10^{-9}$ kg m$^{-2}$ s$^{-1}$) (Figure 3e).

## 4 Discussions and Conclusions

In the carbon-neutral future scenario, reductions in GHGs and aerosols for climate mitigation and environmental improvement could change meteorological conditions and further influence dust emissions and concentrations. However, critical knowledge gaps remain in dust response to future climate change for pursuing carbon neutrality goals. While existing work has captured the combined impacts of anthropogenic aerosols and GHGs on dust flux under different future scenarios (Singh et al., 2017; Woodward et al., 2005; Zhao et al., 2023; Liu et al., 2024), the distinct roles of anthropogenic aerosols versus GHGs in modulating dust flux remain unresolved. Our work systematically resolves these knowledge gaps. In this study, the individual impacts of anthropogenic aerosols and GHGs reductions under the global carbon neutral scenario on dust emissions and concentrations over the dust belt of low- to mid-latitudes in the Northern Hemisphere are investigated using the fully coupled CESM1 model. The distinct effects of future GHGs and aerosol emission changes on dust emissions are individually assessed. Under carbon neutral scenario (SSP1-1.9), significant reductions in dust emissions (3–12%) and concentrations (4–16%) are seen over major Asian and African dust source regions relative to the high fossil fuel scenario (SSP5-8.5) in 2060 (Figures 4a-b).

Anthropogenic aerosols and GHGs reduction exert opposite impacts on dust emissions. Due to aerosol reductions toward carbon neutrality, atmospheric convective is amplified, elevating surface wind speeds by 0.05–0.10 m s$^{-1}$ and intensifying dust emissions (exceeding $2\times10^{-9}$ kg m$^{-2}$ s$^{-1}$) and concentrations (exceeding 30 μg m$^{-3}$), particularly in the North African, Central Asian, South Asian, and East Asian source sectors, by year 2060. Additionally, the reduction in aerosols is expected to increase near-surface temperature by 0.3-1.2°C, decreasing relative humidity and soil water content, further intensifying dust emissions. In contrast, GHGs reduction diminishes the land-ocean thermal contrast, suppressing surface winds by 0.01–0.1 m s$^{-1}$ and associated dust emissions by $2\times10^{-9}$ kg m$^{-2}$ s$^{-1}$ and concentrations by 50 μg m$^{-3}$ in North Africa and Central Asia (Figures 3e-f ). The marked temperature reduction also elevates relative humidity by 1–3%, suppressing dust generation, due to the GHGs reductions. Dust emissions over parts of the Taklamakan Desert and Gobi Desert are promoted, because of a decrease in precipitation and soil water. However, decreases in wind speed enhance dust deposition, leading to a decline in near-surface dust concentrations.

Under combined GHG and aerosol reductions, dust emissions decline by 3%–12%
across Northern Africa and Central Asia, contrasting with an increase of 3%–9% in East
Asia (Figures 4a-b). A consistent pattern has been observed in previous research (Liu
et al., 2024). Correspondingly, surface wind speeds decrease by 0.01–0.1 m/s across
Northern Africa and Central Asia but increase by 0.01–0.05 m/s over East Asia (Figure
12a). Concurrently, relative humidity rises more significantly by 0.1%–3% over major
dust source regions (Figure 12b). This increase raises the wind speed threshold for dust
emission, thereby suppressing dust uplift. However, in East Asia, higher wind speeds
offset the suppression from increased humidity. Changes in soil moisture and
precipitation are insignificant in these dust source regions and thus play minor roles in
dust emission (Figure 12c-d). Consequently, the suppressive effect of GHG mitigation
dominates over the promotive effect of aerosol mitigation in Northern Africa and
Central Asia. This outcome primarily results from the stronger cooling effect induced
by GHG reductions compared to the warming caused by anthropogenic aerosol
reductions (Figure 13a). The cooling diminishes the land–ocean thermal contrast across
Africa and Central Asia, further suppressing wind speeds and inhibiting dust emissions
(Figure 13b-d). In contrast, elevated wind speeds over East Asia are linked to an
intensified Mongolia–Siberian High under joint mitigation, as indicated by sea level
pressure increases of 40–80 Pa in Figure 13e. This enhanced pressure gradient
strengthens surface winds and promotes dust emissions across source regions in East
Asia. This study addresses the critical knowledge gaps about the dust response to future
climate change for pursuing carbon neutrality, providing valuable insights to guide the
establishment of dust prevention measures and strategies in global pursuit of carbon
neutrality.
It is noteworthy that the responses of dust emissions and concentrations to the
GHG and aerosol mitigation are not linear. Adding the individual effects of GHGs and
aerosols together, dust emissions and concentrations show less decreases and even
increases in over the Northern Hemisphere dust belt (Figure S1), compared to the
combined effect of GHG and aerosol mitigation (Figure 3). The differences are likely
associated with nonlinear response of wind fields, including both the wind direction
and wind speed, to the temperature changes induced by GHGs and aerosols, which
could offset each other and ultimately lead to divergent responses in dust emissions and
concentrations.
Dust emissions in the Northern Hemisphere reach a maximum in spring, the
predominant season for dust storm occurrence. Therefore, this study focuses primarily
on dust variations in the spring. Nevertheless, changes in the annual mean dust
emissions are also important. Annual mean dust emission changes are highly consistent
with spring patterns, showing increased emissions from aerosol reductions and
decreased emissions from GHGs mitigation (Figure S2).
Although large model uncertainties exist in the projections of climate response to
anthropogenic forcings, and climate simulated in CESM is relatively more sensitive to
anthropogenic forcings than many other global models (Wang et al., 2023; Ren et al.,
2024), inter-model comparisons nevertheless yield consistent results regarding dust
emissions under the SSP1-1.9 and SSP5-8.5 scenarios. Specifically, many CMIP6
models indicate that GHG and aerosol mitigation reduces dust emissions in Northwest
Africa (Figure S3), similar to the CESM simulation. Under future scenario, potential
variations in tropospheric ozone concentrations may introduce additional complexity,
as ozone can modulate key meteorological drivers as a greenhouse gas (Wang et al,
2023; Gao et al, 2022), which can also regulate dust emission processes. It is reasonable
to speculate that the decline in ozone concentrations under carbon neutrality pathways
would lead to a greater reduction in dust emissions relative to SSP5-8.5 than is currently
estimated in this study, if this factor were accounted for. Also, this study does not
consider the land cover change and the potential future forest expansion (Cramer et al.,
2001; Notaro et al., 2007; Jiang et al., 2011) may weaken the dust changes toward
carbon neutrality, which deserves further investigation in future work. Furthermore, as
evidenced in our model validation, the CESM dust simulations exhibit inherent
limitations, primarily originating from the topographic source function, the dust
emission scheme, coarse spatial and vertical model resolution, and PBL
parameterization (Wu et al., 2020; Lindvall et al., 2012), which collectively contribute
to systematic biases in dust emission flux estimates.
Our findings demonstrate that the carbon neutrality scenario leads to an overall
reduction in dust emissions compared to the high fossil fuel scenario, thereby
alleviating future pressures on dust control policies. These results highlight the
importance of advancing carbon neutrality, which not only achieves climate mitigation
targets but also helps reduce dust pollution. Notably, however, East Asia exhibits
anomalous increases in dust emissions. Therefore, while implementing carbon
neutrality policies, it is essential to additionally strengthen regional measures such as
afforestation and the construction of protective forest belts to further prevent dust
storms.
**Author contributions.** YY designed the research; SY performed the simulations and
analyzed the data. All authors including LR, HW, PW, LC, JJ and HL discussed the
results and wrote the paper.

**Acknowledgments.** The Pacific Northwest National Laboratory is operated for the U.S.
Department of Energy by the Battelle Memorial Institute under contract DE-AC05-
76RLO1830.

**Financial support.** This study was supported by the National Key Research and
Development Program of China (Grant 2024YFF0811400), National Natural Science
Foundation of China (Grant 42475032), Natural Science Foundation of the Jiangsu
Higher Education Institutions of China (Grant 24KJB170007), and Jiangsu Innovation
and Entrepreneurship Team (Grant JSSCTD202346).

**Conflict of Interest.** At least one of the (co-)authors is a member of the editorial board
of ACP.

**Code and data availability.** The dust optical depth for 2020 level can be obtained from
CALIPSO                                       satellite                                       retrievals
(https://search.earthdata.nasa.gov/search/granules?p=C1633034978-LARC_ASDC,
last access:1 June 2025). The CESM model is publicly available at
http://www.cesm.ucar.edu/models/ (last access:1 June 2025). The processed modeling
data are available at https://doi.org/10.5281/zenodo.15478736 (last access:1 June 2025).

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

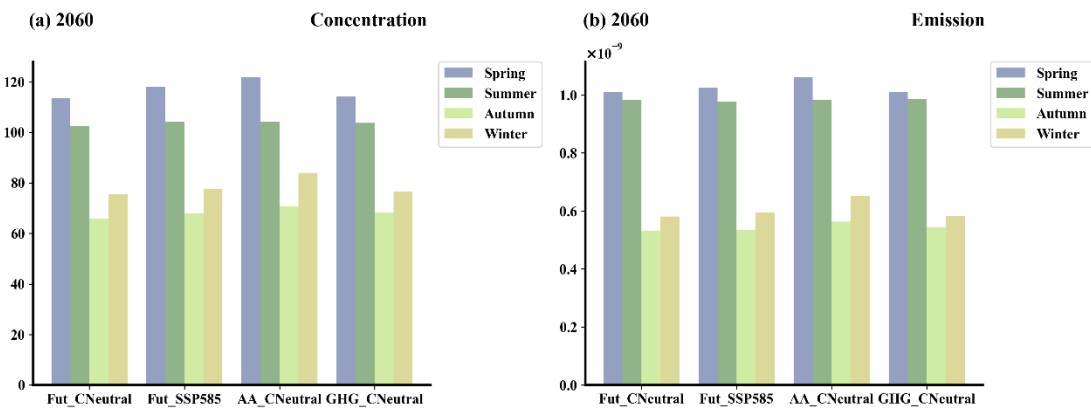

**Figure 1.** Seasonal mean (a) dust near-surface concentration (μg m$^{-3}$) and (b) dust emission (kg m$^{-2}$ s$^{-1}$) during boreal spring (March-April-May), summer (June-July-August), Autumn (September-October-November) and winter (December-January-February) of 2060 over the dust belt (0°–60°N, 25°W–130°E) simulated from the Fut_CNeutral, Fut_SSP585, AA_CNeutral and GHG_CNeutral simulations.

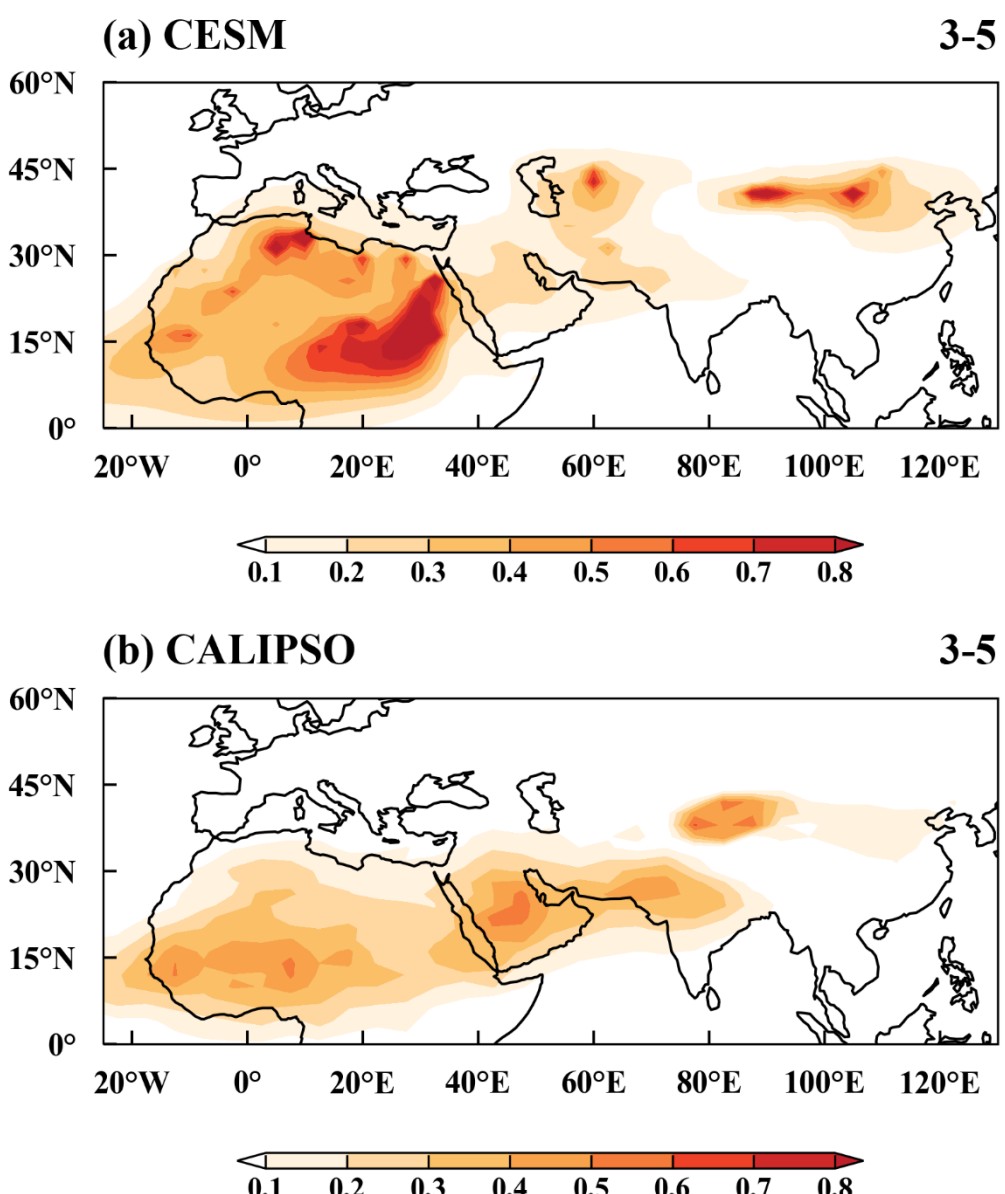

**Figure 2.** Spatial distribution of the average dust optical depth (DOD) from March to May 2020 from (a) the CESM model simulation (Fut_2020) and (b) the CALIPSO satellite observations averaged over 2017–2021.

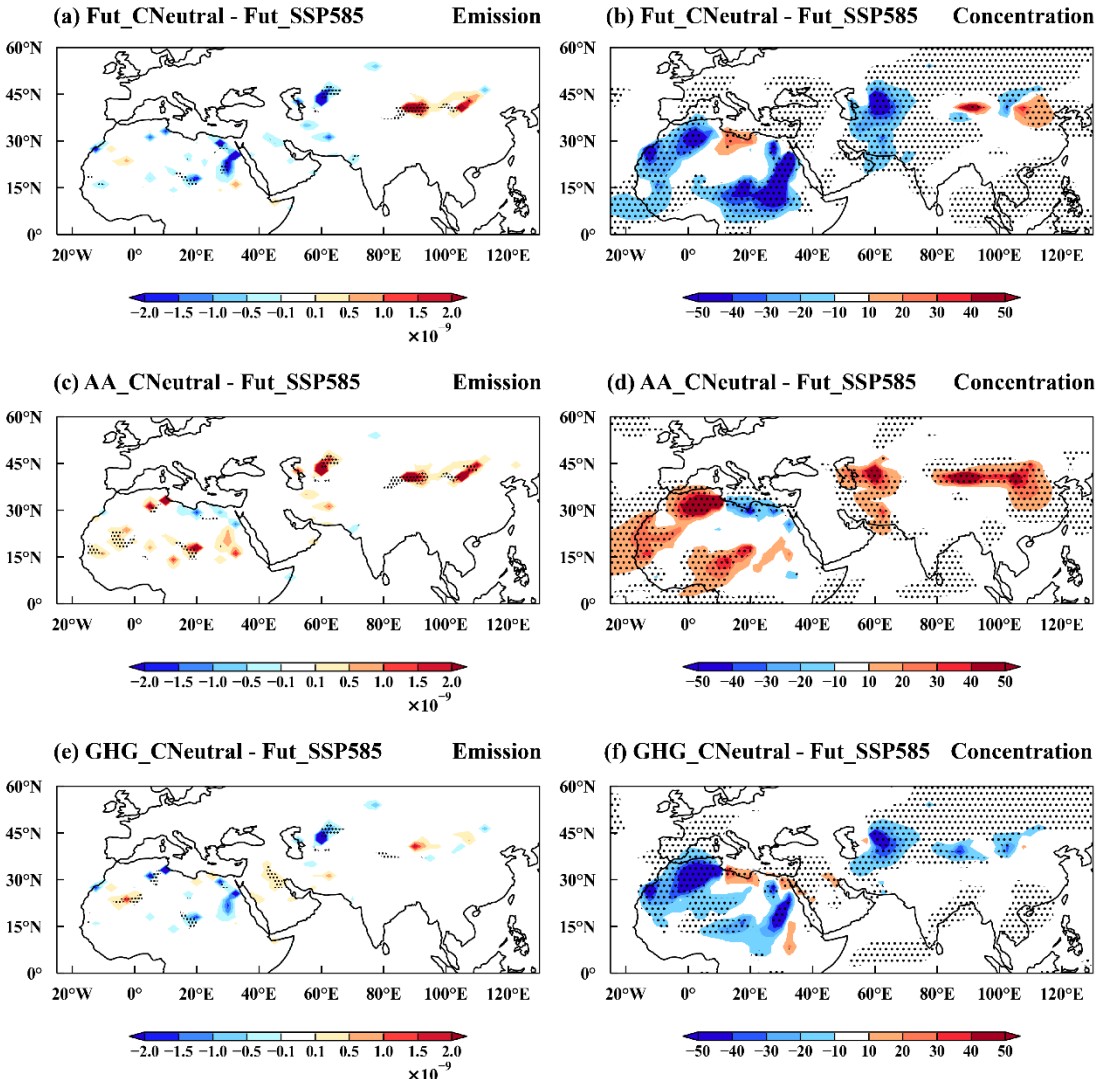

**Figure 3.** Spatial distribution of changes in March–May mean (a, c, e) dust emissions (kg m$^{-2}$ s$^{-1}$) and (b, d, f) near-surface dust concentrations (µg m$^{-3}$) in 2060 for Fut_CNeutral (top), AA_CNeutral (middle), and GHG_CNeutral (bottom) compared to the Fut_SSP585 simulation. The stippled areas indicate statistically significant differences at the 90% confidence level based on a two-tailed Student's t test.

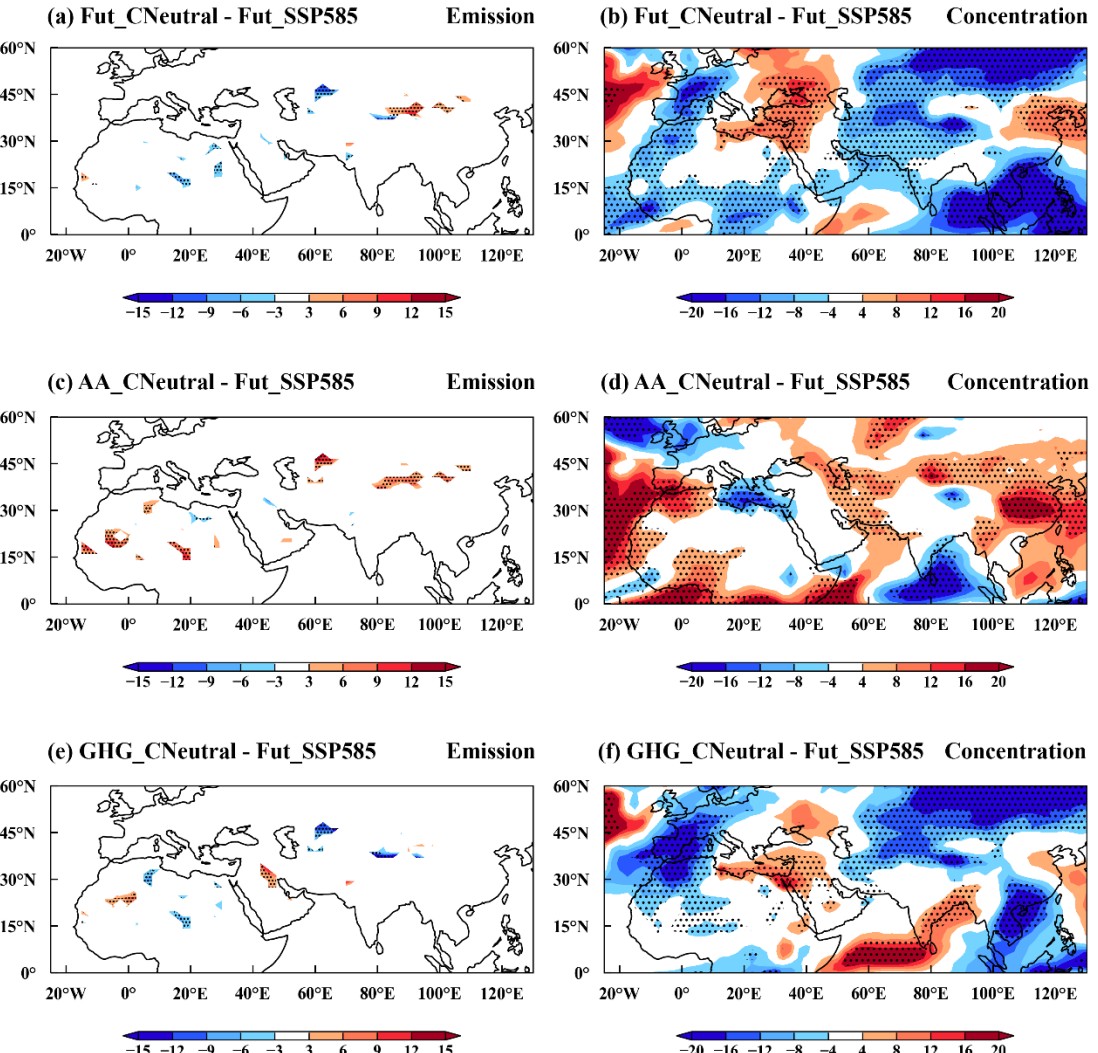

**Figure 4.** Spatial distribution of percentage changes in March–May mean (a, c, e) dust emissions (%) and (b, d, f) near-surface dust concentrations (%) in 2060 for Fut_CNeutral (top), AA_CNeutral (middle), and GHG_CNeutral (bottom) compared to the Fut_SSP585 simulation. The stippled areas indicate statistically significant differences at the 90% confidence level based on a two-tailed Student's t-test.

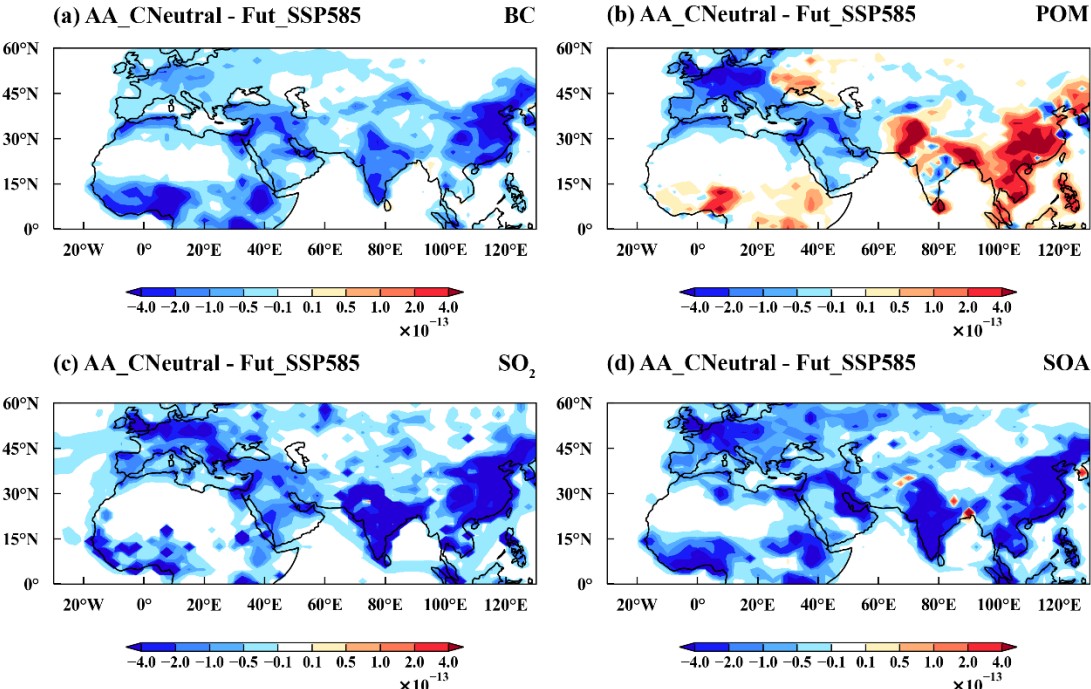

**Figure 5.** Spatial distribution of changes in March–May mean (a) black carbon (BC, kg m$^{-2}$ s$^{-1}$), (b) particulate organic matter (POM, kg m$^{-2}$ s$^{-1}$), (c) sulfur dioxide (SO$_2$, kg m$^{-2}$ s$^{-1}$), and (d) precursor gas of secondary organic aerosol (SOAG, Tg m$^{-2}$ yr$^{-1}$) in 2060 for AA_CNeural, compared to the Fut_SSP585 simulation.

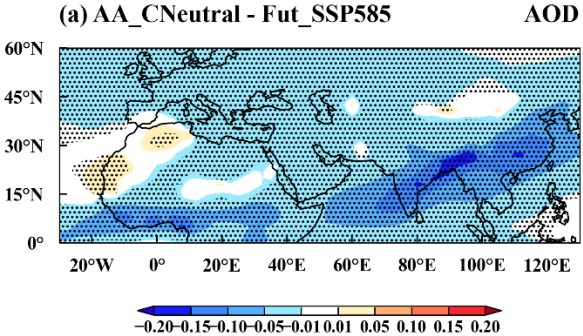

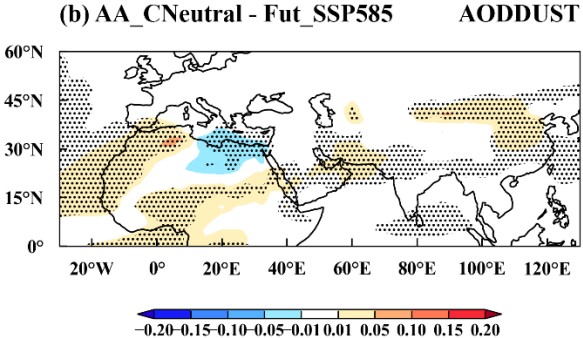

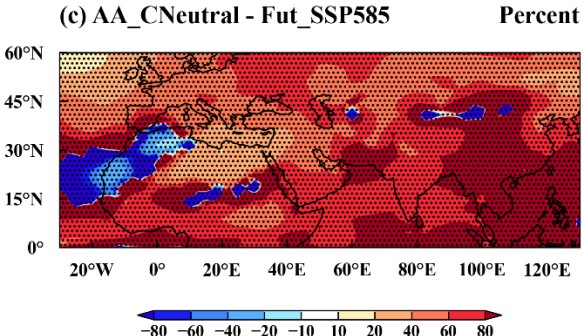

**Figure 6.** Spatial distribution of changes in March–May mean (a) aerosol optical depth (AOD), (b) aerosol optical depth from dust (AODDUST), and (c) the fraction of sulfate AOD change in total AOD change (%) in 2060 for AA_CNeural, compared to the Fut_SSP585 simulation.

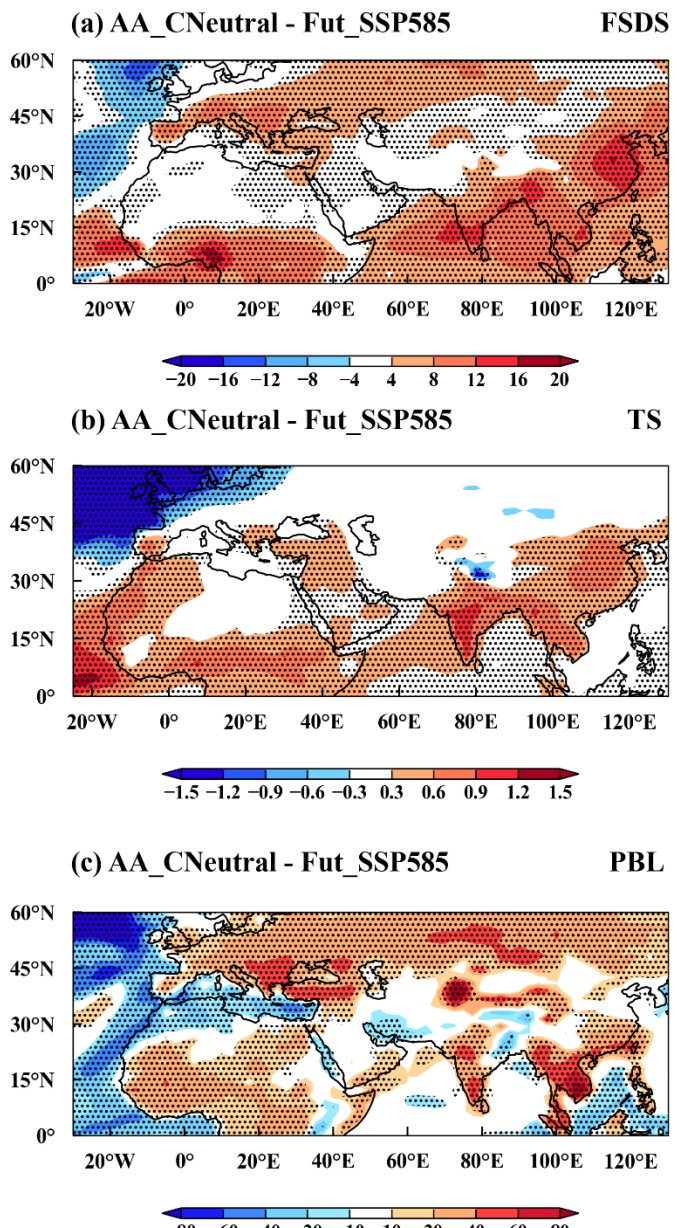


**Figure 7.** Spatial distribution of changes in March–May mean (a) downwelling solar
flux at the surface (FSDS, W/m$^2$), (b) surface temperature (TS, K), and (c) planetary
boundary layer height (PBL, m), in 2060 for AA_CNeural, compared to the
Fut_SSP585 simulation. The stippled areas indicate statistically significant differences
at the 90% confidence level based on a two-tailed Student's t test.

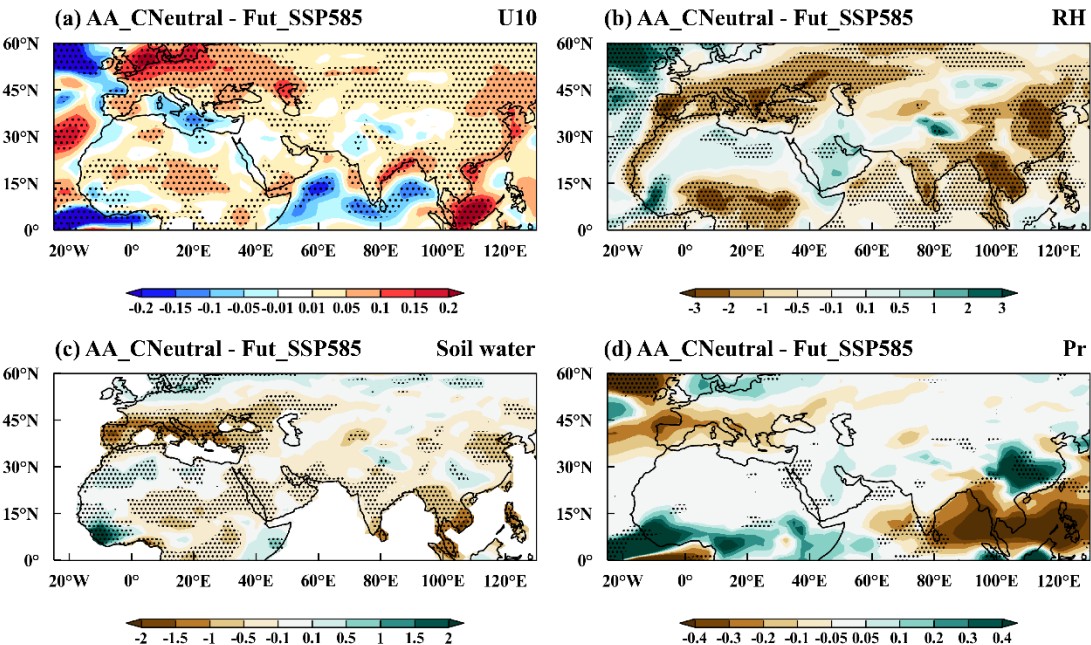


**Figure 8.** Spatial distribution of changes in March–May mean (a) 10-meter wind speed (U10, m s$^{-1}$), (b) relative humidity (RH, %), (c) soil water content (soil water, kg m$^{-2}$), and (d) precipitation rate (pr, mm day$^{-1}$) in 2060 for AA_CNeural, compared to the Fut_SSP585 simulation. The stippled areas indicate statistically significant differences at the 90% confidence level based on a two-tailed Student's t test.

867

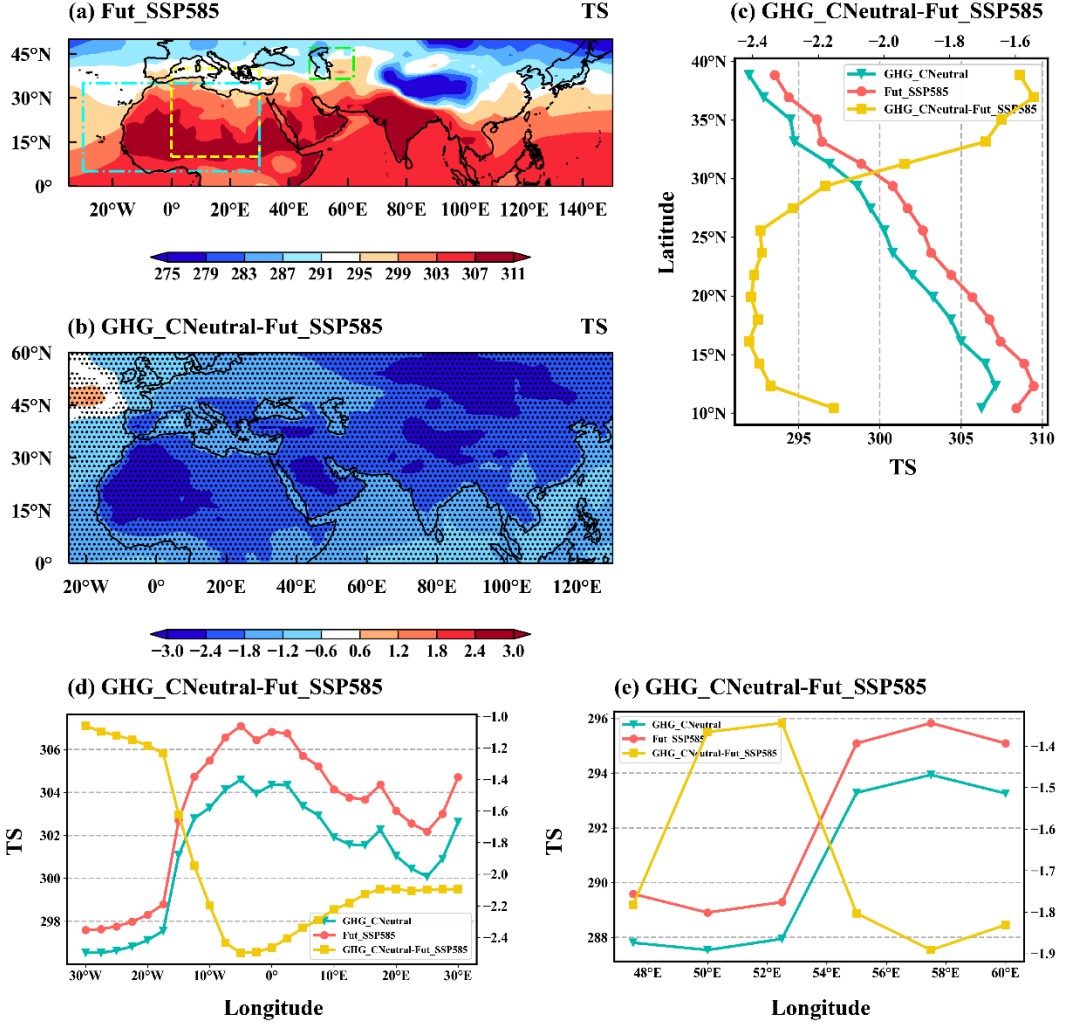

**Figure 9.** Spatial distribution of March–May mean (a) surface temperature (TS, K) in 2060 from Fut_SSP585 and (b) changes in March-May mean surface temperature (TS, K) in 2060 for GHG_CNeural, compared to the Fut_SSP585 simulation. The stippled areas in (a) and (b) indicate statistically significant differences at the 90% confidence level based on a two-tailed Student's t test. (c) Zonal averaged TS (K) over the region (10°–40°N, 0°–30°E, yellow) and (d-e) meridional averaged TS (K) over the regions (5°–35°N, 30°W–30°E, blue; 36.5°–47°N, 47°–62°E, green) marked in (a) from March to May in 2060 for GHG_CNeutral, Fut_SSP5-8.5, and the changes between GHG_CNeutral and Fut_SSP5-8.5.

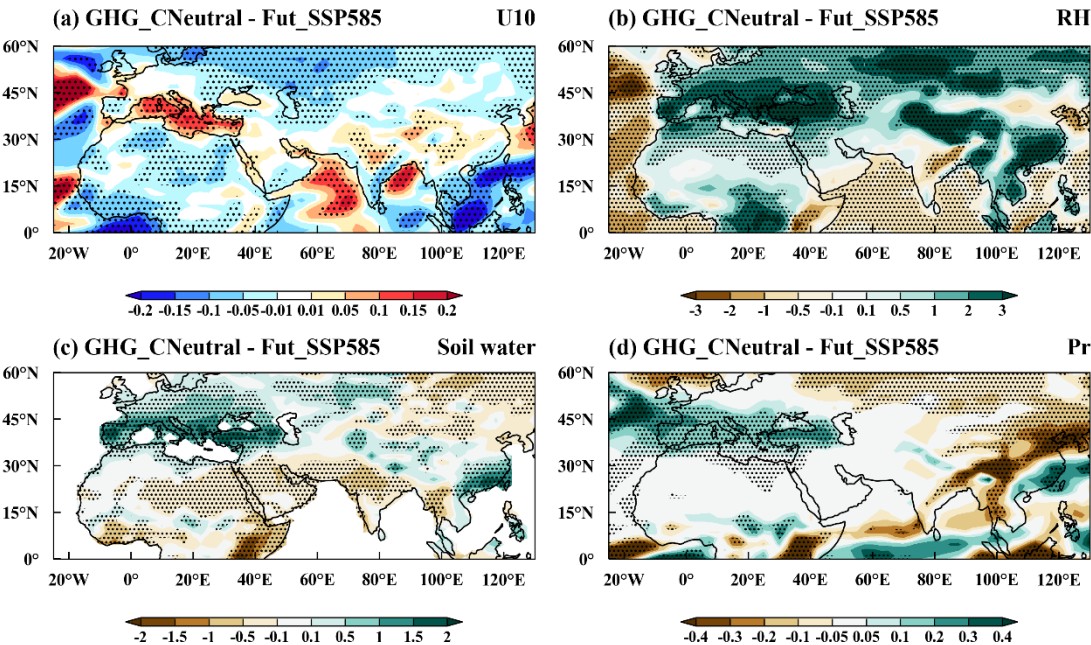

Figure 10. Spatial distribution of changes in March–May mean (a) 10-meter wind speed (U10, m s$^{-1}$), (b) relative humidity (RH, %), (c) soil water content (soil water, kg m$^{-2}$), and (d) precipitation rate (pr, mm day$^{-1}$) in 2060 for GHG_CNeural, compared to the Fut_SSP585 simulation. The stippled areas indicate statistically significant differences at the 90% confidence level based on a two-tailed Student's t test.

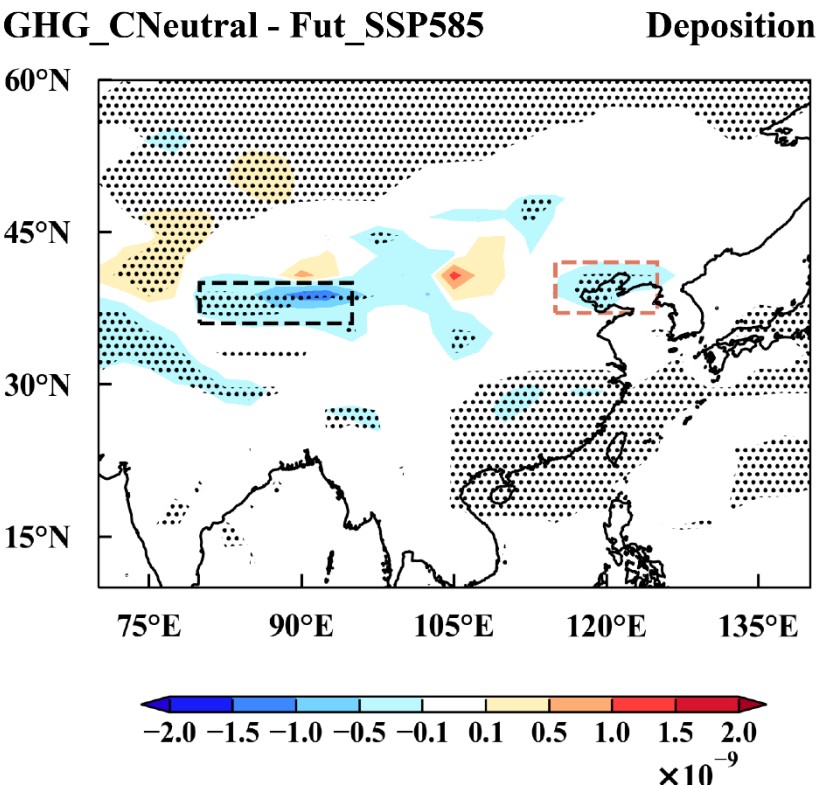

**GHG_CNeutral - Fut_SSP585**     **Deposition**

**Figure 11.** Spatial distribution of dust deposition (kg m$^{-2}$ s$^{-1}$) changes for the period of
March to May in 2060 between GHG_CNeutral and Fut_SSP585 scenarios. The stippled
areas indicate statistically significant differences at the 90% confidence level based on
a two-tailed Student's t test. Negative values denote more dust deposition to the surface.
The Taklimakan (black box) and North China Plain (brown box) are highlighted.

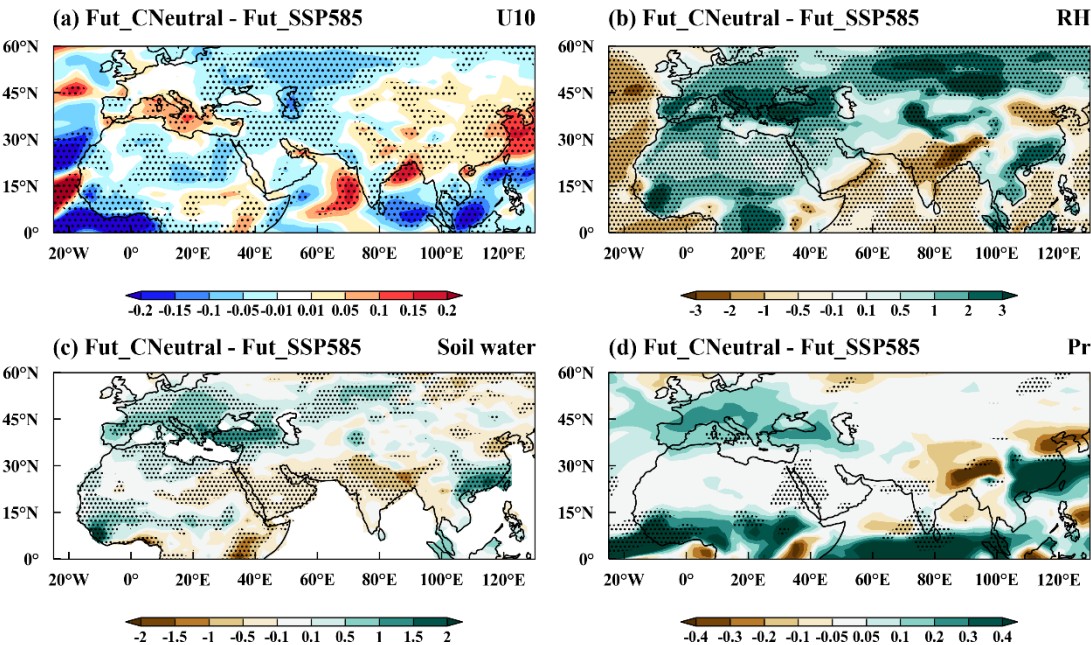


**Figure 12.** Spatial distribution of changes in March–May mean (a) 10-meter wind
speed (U10, m s$^{-1}$), (b) relative humidity (RH, %), (c) soil water content (soil water, kg
m$^{-2}$), and (d) precipitation rate (pr, mm day$^{-1}$) in 2060 for Fut_CNeutral, compared to
the Fut_SSP585 simulation. The stippled areas indicate statistically significant
differences at the 90% confidence level based on a two-tailed Student's t test.

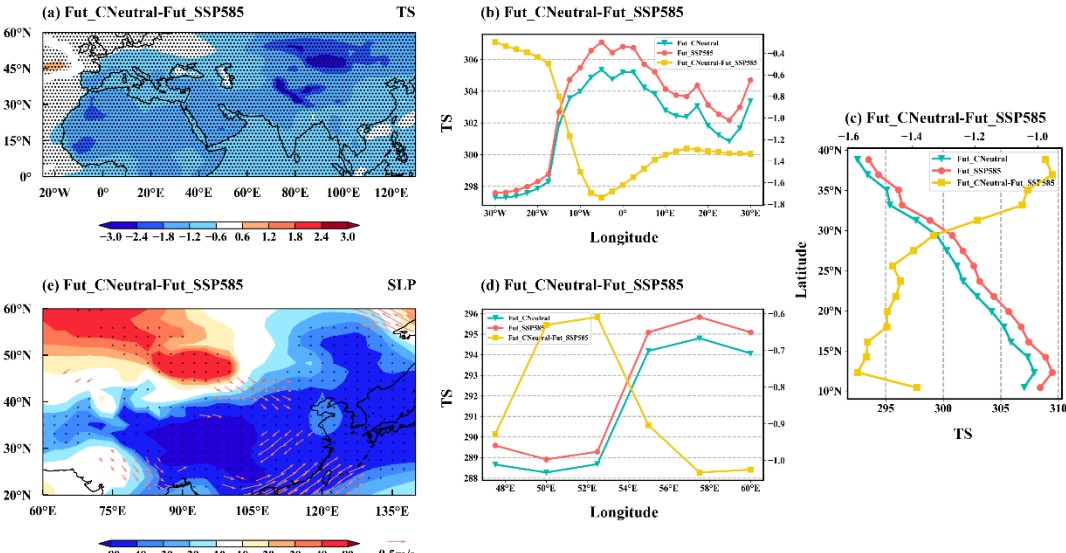


**Figure 13.** Spatial distribution of changes in March–May mean (a) surface temperature
(TS, K) and (e) sea level pressure (SLP, Pa) in 2060 for Fut_CNeutral, compared to the
Fut_SSP585 simulation. The stippled areas in (a) and (b) indicate statistically
significant differences at the 90% confidence level based on a two-tailed Student's t
test. (c) Zonal averaged TS (K) over the region (10°–40°N, 0°–30°E) and (b, d)
meridional averaged TS (K) over the regions (5°–35°N, 30°W–30°E; 36.5°–47°N, 47°–
62°E) marked in (a) from March to May in 2060 for Fut_CNeutral, Fut_SSP5-8.5, and
the changes between Fut_CNeutral and Fut_SSP5-8.5.