# Peer review of "Impacts of reductions in anthropogenic aerosols and greenhouse gases toward carbon neutrality on dust pollution over the Northern Hemisphere dust belt"

_EGUsphere, 2025_

## Referee Comment (RC1)

**Review of "Impacts of reductions in anthropogenic aerosols and greenhouse gases toward carbon neutrality on dust pollution"**

**Summary**

This work evaluates future dust pollution by modeling different climate change scenarios that include changes in GHG and aerosol emissions using a global climate model. By including different SSP scenarios, they are able to assess different mechanisms, especially those that have counteracting effects on dust transport and deposition. The manuscript is well written and organized, but there are some aspects that can be improved prior to publication. Mainly, the novelty is not clear enough, the results are described more qualitatively than quantitatively, and lastly, there are not many comparisons of their results with other works. Below are some specific comments regarding these aspects.

**Specific comments**

- I suggest highlighting the novelties of the paper in the abstract, introduction, and summary. In the introduction, similar studies are mentioned. What are the main differences with those? Do your conclusions agree with all the mentioned studies?

- I suggest evaluating the title. I found it a bit generic, considering that there is a focus on specific regions, and that the selection of carbon neutrality scenarios is quite specific too.

- The abstract has a clear message, but quantitative results could greatly support their statements

- Experiment setup: As mentioned at the end of this Section, the model evolves in time. I don't completely understand when the simulations start and how fine the time resolution is, in terms of the prescribed aerosol concentration. Are all simulations initialized with the same aerosol conditions and only the emissions change in time (hourly?) and space, or do they also have different initial conditions? You can also tell us a bit more about how realistic the aerosol emissions are modeled, to better understand the simulation of these scenarios. Are anthropogenic emissions properly specified per region/country/city, and do they vary along the day? Another thing that I wonder is if the model is able to capture changes in vegetation, since it was mentioned as a possible agent in aerosol modification in the Introduction.

- On model evaluation, are the comparisons with CALIPSO performed along a whole year or a specific timeframe, and is it for the whole domain or a region of the space?

- How fine is the vertical resolution in order to observe PBL rising? If not fine enough, could this be a bias that enhances the surface wind strengthening?

- The mechanisms that drive the observed changes are carefully explained, especially those with opposite trends, which seems to be the main strength of this work. However, these changes are not quantitatively described nor compared with other studies. For instance, in L348 "significant reductions" could be quantified in a comparative way (as a % of the initial scenario, for example), in order to have a more complete description of their results.

- Fig. 1: "autumn"

---

## Author Comment (AC1)

This work evaluates future dust pollution by modeling different climate change scenarios that include changes in GHG and aerosol emissions using a global climate model. By including different SSP scenarios, they are able to assess different mechanisms, especially those that have counteracting effects on dust transport and deposition. The manuscript is well written and organized, but there are some aspects that can be improved prior to publication. Mainly, the novelty is not clear enough, the results are described more qualitatively than quantitatively, and lastly, there are not many comparisons of their results with other works. Below are some specific comments regarding these aspects.

We thank the reviewer for the constructive comments and suggestions, which are very helpful for improving the clarity and reliability of the manuscript. Please see our point-by-point responses to your comments below.

1) I suggest highlighting the novelties of the paper in the abstract, introduction, and summary. In the introduction, similar studies are mentioned. What are the main differences with those? Do your conclusions agree with all the mentioned studies?

**Reply:** The novelties of this study have been highlighted in the Abstract (L32-34), Introduction (L132-135, L162-173, L166-171), and Conclusion (L430-438).

Existing studies typically focus on dust flux responses to climate change under future scenarios, thereby examining only the combined effects of anthropogenic aerosols and GHGs, which also have yet to quantify dust response to future climate change for pursuing carbon neutrality goals (Zhao et al., 2023; Liu et al., 2024). In this study, the individual impacts of anthropogenic aerosols and GHGs reductions under carbon neutral scenario on dust emissions and concentrations over the dust belt of low- to midlatitudes in the Northern Hemisphere are investigated.

Our conclusions agree with all the mentioned studies. Dust emissions are significantly higher under high-emission scenarios than under low-emission pathways (e.g., Singh et al., 2017; Zhao et al., 2023; Liu et al., 2024; Gomez et al., 2023), consistent with our finding of reduced dust emissions in carbon neutral scenario relative to the high fossil fuel scenario. Moreover, the impacts of anthropogenic aerosol and GHGs mitigation on wind speed identified in this study are in accordance with previous findings (e.g., Lei et al., 2023; Ren et al., 2024; Sawadogo et al., 2019).

2) I suggest evaluating the title. I found it a bit generic, considering that there is a focus on specific regions, and that the selection of carbon neutrality scenarios is quite specific too.

**Reply:** We have now revised the title to "Impacts of reductions in anthropogenic aerosols and greenhouse gases toward carbon neutrality on dust pollution over the Northern Hemisphere dust belt". Considering that SSP1-1.9 has been widely used as the carbon neutrality scenario, it is not specified in the title.

3) The abstract has a clear message, but quantitative results could greatly support their statements.

**Reply:** Quantitative results have been added (L40-44) in abstract. For example, (i) Reductions in aerosols amplify surface downwelling shortwave radiation, convection and wind speed, thereby promoting dust emissions by 6–12% and concentrations by 4–20% over North Africa, the Central Asia Desert and East Asia; (ii) GHGs reductions diminish the land-ocean thermal contrast and wind speed, suppressing dust emissions by 6–15% and concentrations by 8–20% mainly over the Central Asia Desert and North Africa.

4) Experiment setup: As mentioned at the end of this Section, the model evolves in time. I don't completely understand when the simulations start and how fine the time resolution is, in terms of the prescribed aerosol concentration. Are all simulations initialized with the same aerosol conditions and only the emissions change in time (hourly?) and space, or do they also have different initial conditions? You can also tell us a bit more about how realistic the aerosol emissions are modeled, to better understand the simulation of these scenarios. Are anthropogenic emissions properly specified per region/country/city, and do they vary along the day? Another thing that I wonder is if the model is able to capture changes in vegetation, since it was mentioned as a possible agent in aerosol modification in the Introduction.

**Reply:** Equilibrium simulations are run for 100 years of the year 2060, with the initial conditions at the year 2060 level. The initial 40 years are considered as model spin-up period. The output data has a monthly temporal resolution. All simulations are initialized with the same aerosol and GHG conditions and only the aerosol emissions and/or GHGs concentrations change in time and space every month. The model is integrated every 30 minutes and the results are archived every month. Future emission inventories build on the Shared Socioeconomic Pathways, providing standardized multidimensional parameters (e.g., population, economy, technology, environment, institutions) and qualitative narratives at national/regional scales (van Vuuren et al., 2017; Kriegler et al., 2017; Fujimori et al., 2017; Calvin et al., 2017; Fricko et al., 2017).

This study aims to investigate the influence of meteorological factors on dust emission under future climate changes. In the model simulations, land use is held constant, thereby unable to account for potential vegetation changes. However, based on previous studies, we can reasonably assume the impact of vegetation dynamics on dust emissions. Notaro et al. (2006) employed a fully coupled atmosphere–ocean–land–ice model with dynamic vegetation to analyze future vegetation changes under continuously increasing $CO_2$ concentrations. Their results revealed an increase in tree cover across arid regions, such as the Sahel and the Middle East, along with a northward shift of the Sahel

transition zone. Cramer et al. (2001) demonstrated that the physiological effect can facilitate forest expansion into savanna and grassland expansion into arid tropical regions. Furthermore, by using an asynchronously coupled system between the IAP-AGCM model and the biosphere BIOME3 model, Jiang et al. (2011) projected an increase in deciduous forests across tropical Africa under the A2 emissions scenario. Consequently, the vegetation changes may weaken the dust changes in the future. We have now added it in the manuscript.

5) On model evaluation, are the comparisons with CALIPSO performed along a whole year or a specific timeframe, and is it for the whole domain or a region of the space?

**Reply:** For model evaluation, CALIPSO satellite observations are compared against simulations across the entire study domain (0°–60°N, 25°W–130°E) during March-May of 2017－2021, since that the data end in 2021.

6) How fine is the vertical resolution in order to observe PBL rising? If not fine enough, could this be a bias that enhances the surface wind strengthening?

**Reply:** The model has 30 vertical layers, from the surface to the top of the atmosphere. however, this resolution remains relatively low. Lindvall et al. (2012) evaluated the performance of PBL parameterizations in CESM using observations and reanalysis data across a range of near-surface parameters. Their results indicate that the model captures spatial patterns relatively well but systematically underestimates PBL height. Consequently, this simulated PBL bias may influence wind speed changes. We have now added the discussion about this potential bias.

7) The mechanisms that drive the observed changes are carefully explained, especially those with opposite trends, which seems to be the main strength of this work. However, these changes are not quantitatively described nor compared with other studies. For instance, in L348 "significant reductions" could be quantified in a

comparative way (as a % of the initial scenario, for example), in order to have a more complete description of their results.

**Reply:** Quantitative descriptions of dust flux changes and key meteorological drivers (e.g., wind speed) have been added to Section 3 and Section 4. For example, anthropogenic aerosol reductions in SSP1-1.9 relative to SSP5-8.5 amplify 10-m wind speed by 0.05–0.10 m s$^{-1}$ across core dust sources (Figures 8a), driving intensified dust emission fluxes by 6–12% and near-surface concentrations by 8–16% in North and Central Africa (Figures 3c-d, Figures 4c-d). The GHGs reduction elevates relative humidity (Figure 10b), which raises the critical threshold wind velocity required for dust mobilization. It further reduces dust emission fluxes by 6–15% and atmospheric dust concentrations by 8–20% (Figure 4e-f), particularly in the North African and Central Asian source regions, even though the soil moisture slightly increases in some regions (Figure 10c).

[revised manuscript text omitted]

---

## Author Comment (AC2)

**Response to Reviewer 2**

This study investigates an interesting and underexplored aspect of climate policy: the unintended consequences of pursuing carbon neutrality on mineral dust pollution. Using the fully coupled Community Earth System Model (CESM1), the authors conduct a set of sensitivity experiments to investigate the individual and combined impacts of reducing greenhouse gases (GHGs) and aerosols under a carbon-neutral scenario versus a high-emission scenario on future dust emissions and concentrations. The authors conclude that GHG reductions and their associated dust-suppressing effect dominate the overall response, offsetting the dust increase caused by aerosols. This highlights important implications of decarbonization beyond improvements in air quality and emphasizes the complex and competing geophysical feedbacks. The study is interesting and relevant. However, I feel the authors should address several comments before full acceptance.

We thank the reviewer for the constructive comments and suggestions, which are very helpful for improving the clarity and reliability of the manuscript. Please see our point-by-point responses to your comments below.

**Major comments:**

1) Introduction: somewhat fragmented, I believe it would benefit from some re-organisation. Also, it would be good to highlight and organise by mechanisms/phenomena. The novelty of this study and the gaps need to be better and more clearly presented.

**Reply:** The introduction is organized as follows: 1. Introduction of dust aerosol; 2. Dust source across the globe; 3. Climate influences dust distribution and mechanisms; 4. Dust variation under future climate change; 5. Changing future climate toward carbon neutrality; 6. The objective of this study. We have now slightly revised the first key sentence in each paragraph to emphasize the keynote.

We have now clarified the novelty in many parts in Introduction, such as "However, Existing studies typically focus on dust flux responses to climate change under future scenarios, thereby examining only the combined effects of anthropogenic aerosols and GHGs, which also have yet to quantify dust response to future climate change due to individual changes in anthropogenic aerosols and GHGs for pursuing carbon neutrality goals (Zhao et al., 2023; Liu et al., 2024). In this study, we conduct Earth system model experiments to assess the impact of aerosols and GHGs reductions toward carbon neutrality on meteorological variables such as precipitation, relative humidity, and wind speed, as well as their implications for dust emissions and concentrations."

2)  Changes in vegetation are not mentioned, yet I presume they are included and differ between the future scenarios, and they are expected to exert a significant influence on dust emissions.

**Reply:** land use is held constant, thereby unable to account for potential vegetation changes. However, based on previous studies, we can reasonably assume the impact of vegetation dynamics on dust emissions. Notaro et al. (2006) employed a fully coupled atmosphere–ocean–land–ice model with dynamic vegetation to analyze future vegetation changes under continuously increasing $CO_2$ concentrations. Their results revealed an increase in tree cover across arid regions, such as the Sahel and the Middle East, along with a northward shift of the Sahel transition zone. Cramer et al. (2001) demonstrated that the physiological effect can facilitate forest expansion into savanna and grassland expansion into arid tropical regions. Furthermore, by using an asynchronously coupled system between the IAP-AGCM model and the biosphere BIOME3 model, Jiang et al. (2011) projected an increase in deciduous forests across tropical Africa under the A2 emissions scenario. Consequently, the vegetation changes may weaken the dust changes in the future. We have now added it in the manuscript..

3)  Section 2.3: The comparison between the model's multi-year climatology and 1

year observations is somewhat unfair and biased. The authors should use a multi-year observational record.

**Reply:** Thanks for the suggestion. CALIPSO satellite data averaged over March-May during 2017–2021 has now been used to validate the model performance.

[Figure]

**Figure 2.** Spatial distribution of the average dust optical depth (DOD) from March to May 2020 from (a) the CESM model simulation (Fut_2020) and (b) the CALIPSO satellite observations averaged over 2017–2021.

4) While the authors focus on March-May as this is the season when the largest dust emissions occur, I believe it is even more important to display and describe annual-mean changes, as some effects may partially compensate for the annual mean. Annual means are also more directly placed into the context of global warming, etc. Some other mechanisms are also at play in other seasons. I also wonder whether the prevalence of the GHG-related signal would occur in other seasons,

for example JJA, when we expect a strong Asian monsoon response to regional aerosol changes.

**Reply:** Despite interannual climate changes driven by reductions in anthropogenic aerosols and GHGs, dust emissions in the Northern Hemisphere reach a maximum in spring, the predominant season for dust storm occurrence. Therefore, this study focuses primarily on dust variations in the spring. Nevertheless, changes in the annual mean dust emissions are also important. Annual mean dust emission changes are highly consistent with spring patterns, showing increased emissions from aerosol reductions and decreased emissions from GHGs mitigation (Figure S2). We have now added these discussions in the manuscript.

[Figure]

**Figure S2.** Spatial distribution of changes in annual mean dust emissions (kg m$^{-2}$ s$^{-1}$) in 2060 for Fut_CNeutral (top), AA_CNeutral (middle), and GHG_CNeutral (bottom)

compared to the Fut_SSP585 simulation. The stippled areas indicate statistically significant differences at the 90% confidence level based on a two-tailed Student's t test.

5) The mechanistic analysis is overall sound but has also some points that need to be better and more clearly investigated. For example, some changes displayed in Fig 5 are not collocated with dust emissions (e.g., PBL changes). Aerosol emission changes are mostly over Asia, leading to large regional increases in surface radiation and temperature. How is the signal propagating to remote areas such as North Africa? How is surface wind changing in remote areas? Similarly, how can the authors explain the change in surface wind (not necessarily meridional only) by variations in the meridional temperature gradient under GHG forcing? More physical and dynamical insights are needed.

**Reply:** The spatial patterns of planetary boundary layer (PBL) height changes in Fig. 7 show a mismatch with dust emission changes in some regions. This discrepancy arises from the imperfect correspondence between boundary layer height and surface wind speed. Similar discrepancies have been documented in previous research. For example, Qin et al. (2024), while examining the co-benefits of China's carbon neutrality target, proposed PBL height changes as a mechanism underlying wind speed variations, yet also observed regional inconsistencies between PBL height and surface wind trends. Investigating mechanisms behind aerosol induced reductions in near surface wind speeds, Jacobson et al. (2006) found that aerosol particles enhanced atmospheric stability and diminished turbulent kinetic energy by absorbing and scattering solar radiation. This suppression of vertical momentum exchange reduced the downward mixing of higher-speed winds, leading to lower surface wind speeds. However, they also reported that aerosol increases could enhance winds in some regions, primarily due to aerosol-driven local temperature adjustments modifying pressure gradients. The wind speed response to aerosol changes reported in these studies agrees with our findings, and our mechanistic interpretation that aerosol reduction increases wind speed is also consistent with their established physical understanding. We have now added

the corresponding description in the manuscript.

Anthropogenic aerosol emission changes are primarily concentrated in Asia. However, significant reductions in aerosol optical depth (AOD) are also evident over remote regions including Northern Africa (Figure 6a). A comparison of Figure 6a and 6b indicates that the AOD reduction over most of Northern Africa is not driven by dust aerosols. Rather, sulfate AOD changes dominate the AOD decreases over most of the region, accounting for over 60% of the total reduction, with contributions reaching 20%–40% in some areas (Figure 6c). This finding points to a combination of reduced local anthropogenic emissions as the cause of the AOD decline over remote regions such as Northern Africa. Therefore, the pronounced increase in surface radiation and temperature over Northern Africa and other remote areas cannot be attributed to propagation from Asia. In these regions, anthropogenic aerosol reductions directly lead to the AOD decrease. Now added in the manuscript.

By reducing the land-ocean thermal contrast, GHG mitigation lowers surface wind speeds over major dust source regions, leading to a consequent decline in dust emissions. Previous studies have explored the influence of meridional temperature gradients on wind speed. Qu et al. (2025) studied prolonged wind droughts in a warming climate. Under the SSP5-8.5 scenario, they found that wind droughts decrease in the tropics, primarily due to increased wind speeds. In contrast, northern mid-latitudes experienced more frequent wind droughts, substantially driven by reduced wind speeds. This discrepancy is attributable to contrasting mechanisms. In the tropics, global warming amplifies the land-ocean thermal contrast, thereby strengthening winds. In the mid-latitudes, it reduces meridional temperature gradients, weakening baroclinicity and storm-track activity. Thus, the mechanism of wind speed reduction via GHG-induced diminishment of the land-ocean thermal contrast is consistent with established understanding. Now added in the manuscript.

[Figure]

**Figure 6.** Spatial distribution of changes in March–May mean (a) aerosol optical depth (AOD), (b) aerosol optical depth from dust (AODDUST), and (c) the fraction of sulfate AOD change in total AOD change (%) in 2060 for AA_CNeural, compared to the Fut_SSP585 simulation.

6)  It would be interesting to expand the discussion on linearities (or lack of) in the combined GHG+AER response.

**Reply:** It is noteworthy that the responses of dust emissions and concentrations to the GHG and aerosol mitigation are not linear. Adding the individual effects of GHGs and aerosols together, dust emissions and concentrations show less decreases and even

increases in over the Northern Hemisphere dust belt (Figure S1), compared to the combined effect of GHG and aerosol mitigation (Figure 3). The differences are likely associated with nonlinear response of wind fields, including both the wind direction and wind speed, to the temperature changes induced by GHGs and aerosols, which could offset each other and ultimately lead to divergent responses in dust emissions and concentrations.

[Figure]

**Figure S1.** The sum of individual effects of GHG and aerosol mitigation on the changes in March–May mean (a) dust emissions (kg m$^{-2}$ s$^{-1}$) and (b) near-surface dust concentrations (μg m$^{-3}$) in 2060 (AA_CNeutral – Fut_SSP585 + GHG_CNeutral – Fut_SSP585).

7)  Finally, results should be more extensively discussed in the context of existing studies. Can the authors infer some qualitative conclusions on what would happen with other CMIP6 models (even if these specific experiments do not exist) with different climate sensitivities and aerosol radiative forcing? Do we expect GHG to dominate as well?

**Reply:** Inter-model comparisons of CMIP6 simulations under the SSP1-1.9 and SSP5-8.5 scenarios reveal certain inter-model discrepancies in future dust emission projections (Figure S3). Nevertheless, GHG and aerosol mitigation reduces dust emissions in Northwest Africa, as indicated by the majority CMIP6 models and the CESM simulation. Now added in the manuscript.

[Figure]

**Figure S3.** Spatial distribution of the March–May mean dust emission change (kg m$^{-2}$ s$^{-1}$) between the SSP1-1.9 and SSP5-8.5 scenarios from (a–g) individual CMIP6 models and (h) the multi-model mean. The stippled areas indicate statistically significant differences at the 90% confidence level based on a two-tailed Student's t test.

**Minor comments:**

8)  The tile is not fully clear, particularly "toward carbon neutrality".

**Reply:** The SSP1-1.9 has been widely used to represent the carbon neutrality scenario

(e.g., Wang et al., 2023; Zhu et al., 2024). In the abstract, we have specified it referred to SSP1-1.9 scenario.

9) Introduction: I would replace the word "demonstrated" with found, showed, etc.

**Reply:** Thanks. Modified to "found/showed".

10) L57: "Dust have been demonstrated", rephrase

**Reply:** Thanks. Modified to "Dust can".

11) L76: Is the decreased warming continuing to present day? As the paragraph is discussing the effects of anthropogenic factors, this sentence is a bit disconnected from the broader context.

**Reply:** The warming hiatus occurred between 2002 and 2014 but did not persist beyond this period (Jin et al., 2017). We have now removed this sentence from the text.

12) L81: Regarding the link between GHG and the NAO, is that also valid in the future?

**Reply:** This is one possible linkage of GHG, NAO and dust transport to South Asia. However, the dust transport can be affected by many other factors. This is why the model does not simulate a significant change in dust loading in South Asia in the future.

13) L86: "global warming induced surface warming" redundant

**Reply:** Corrected.

14) L90: add some context concerning the period

**Reply:** Added as "Analyses of observations from 1979 to 2013".

15) L146-156: confusing

**Reply:** We have now removed this confusing sentence.

16) Section 2.2: a brief description of why ssp119 is carbon neutral and why the authors

chose 2060 as the reference year.

**Reply:** The SSP1-1.9 represents a sustainable development scenario focused on ecological restoration, conservation, and a significant reduction in fossil fuel dependence. This pathway is considered the most likely to achieve the 1.5 °C target under the Paris Agreement and carbon neutrality in the mid-21st century (Su et al., 2021; Wang et al., 2023; Zhu et al., 2024).

This study aims to assess the impact of carbon neutrality targets on dust climate. Many countries had committed to achieving carbon neutrality by 2050 or 2060, with most targets set for the post-2050 period (Chen et al., 2022). Focusing on the year 2060 therefore ensures direct alignment with policy timelines and enhances the practical relevance of our results.

17) L243: present the …
**Reply:** Modified to "present the".

18) L245: they are not time-varying, they are fixed at 2060 values
**Reply:** Changed to "fixed anthropogenic aerosols and GHGs".

19) L330: monsoon? The monsoon season in summer…
**Reply:** Changed to "atmospheric circulation".

**References**

Chen, L., Msigwa, G., Yang, M., Osman, A. I., Fawzy, S., Rooney, D. W., and Yap, P-S.: Strategies to achieve a carbon neutral society: a review, Environ. Chem. Lett., 20, 2277–2310, https://doi.org/10.1007/s10311-022-01435-8, 2022.

Cramer, W., Bondeau, A., Woodward, F. I., Prentice, I. C., Betts, R. A., Brovkin, V., Cox, P. M., Fisher, V., Foley, J. A., Friend, A. D., Kucharik, C., Lomas, M. R., Ramankutty, N., Sitch, S., Smith, B., White, A., and Young-Molling, C.: Global response of terrestrial ecosystem structure and function to CO2 and climate change: Results from six dynamic global vegetation models, Global. Change Biology, 7(4), 357-373, https://doi.org/10.1046/j.1365-2486.2001.00383.x, 2001.

Hu, Z. Z., and Wu, Z.: The intensification and shift of the annual North Atlantic Oscillation in a global warming scenario simulation, Tellus A, 56(2), 112–124, https://doi.org/10.3402/tellusa.v56i2.14403, 2004.

Jacobson, M. Z., and Kaufman, Y. J.: Wind reduction by aerosol particles, Geophys. Res. Lett., 33, L24814, https://doi.org/10.1029/2006GL027838, 2006.

Jiang, D., Zhang, Y. and Lang, X.: Vegetation feedback under future global warming, Theor. Appl. Climatol., 106, 211–227, https://doi.org/10.1007/s00704-011-0428-6, 2011.

Jin, Q., and Wang, C.: A revival of Indian summer monsoon rainfall since 2002, Nat. Clim. Change, 7, 587–594, https://doi.org/10.1038/nclimate3348, 2017.

Liu, J., Wang, X., Wu, D., Wei, H., Li, Y., and Ji, M.: Historical footprints and future projections of global dust burden from bias-corrected CMIP6 models, npj Clim. Atmos. Sci., 7, 1, https://doi.org/10.1038/s41612-023-00550-9, 2024.

Notaro, M., Vavrus, S., and Liu, Z.: Global Vegetation and Climate Change due to Future Increases in CO2 as Projected by a Fully Coupled Model with Dynamic Vegetation, J. Climate, 20, 70–90, https://doi.org/10.1175/JCLI3989.1, 2007.

Qin, Y., Zhou, M., Hao, Y., Huang, X., Tong, D., Huang, L., Zhang, C., Cheng, J., Gu, W., Wang, L., He, X., Zhou, D., Chen, Q., Ding, A., and Zhu, T.: Amplified positive effects on air quality, health, and renewable energy under China's carbon

neutral target, Nat. Geosci., 17, 411–418, https://doi.org/10.1038/s41561-024-01425-1, 2024.

Qu, M., Shen, L., Zeng, Z., Yang, B., Zhong, H., Yang, X. and Liu, X.: Prolonged wind droughts in a warming climate threaten global wind power security, Nat. Clim. Chang. 15, 842–849, https://doi.org/10.1038/s41558-025-02387-x, 2025.

Singh, C., Ganguly, D., and Dash, S. K.: Dust load and rainfall characteristics and their relationship over the South Asian monsoon region under various warming scenarios, J. Geophys. Res. Atmos., 122, 7896-7921, https://doi.org/10.1002/2017JD027451, 2017.

Su, B., Huang, J., Mondal, S. K., Zhai, J., Wang, Y., Wen, S., Gao, M., Lv, Y., Jiang, S., Jiang, T., and Li, A.: Insight from CMIP6 SSP-RCP scenarios for future drought characteristics in China, Atmos. Res., 250, 105375, https://doi.org/10.1016/j.atmosres.2020.105375., 2021.

Wang, P., Yang, Y., Xue, D., Ren, L., Tang, J., Leung, L. R., and Liao, H.: Aerosols overtake greenhouse gases causing a warmer climate and more weather extremes toward carbon neutrality, Nat. Commun., 14, 7257, https://doi.org/10.1038/s41467-023-42891-2, 2023.

Zhao, Y., Yue, X., Cao, Y., Zhu, J., Tian, C., Zhou, H., Chen, Y., Hu, Y., Fu, W., and Zhao, X.: Multi-model ensemble projection of the global dust cycle by the end of 21st century using the Coupled Model Intercomparison Project version 6 data, Atmos. Chem. Phys., 23, 7823–7838, https://doi.org/10.5194/acp-23-7823-2023, 2023.

Zhu, J., Yang, Y., Wang, H., Gao, J., Liu, C., Wang, P., and Liao, H., Impacts of projected changes in sea surface temperature on ozone pollution in China toward carbon neutrality, Sci. Total Environ., 915, 170024, https://doi.org/10.1016/j.scitotenv.2024.170024, 2024.